# Quantifying the Effects of COVID-19 on Mental Health Support Forums

**Laura Biester**[*]**, Katie Matton**[*]**, Janarthanan Rajendran,**
**Emily Mower Provost, Rada Mihalcea**
Computer Science & Engineering, University of Michigan, USA
`{lbiester,katiemat,rjana,emilykmp,mihalcea}@umich.edu`

## Abstract

The COVID-19 pandemic, like many of the disease outbreaks that have preceded it, is likely to have a profound effect on mental health. Understanding its impact can inform strategies for mitigating negative consequences. In this work, we seek to better understand the effects of COVID-19 on mental health by examining discussions within mental health support communities on Reddit. First, we quantify the rate at which COVID-19 is discussed in each community, or subreddit, in order to understand levels of pandemic-related discussion. Next, we examine the volume of activity in order to determine whether the number of people discussing mental health has risen. Finally, we analyze how COVID-19 has influenced language use and topics of discussion within each subreddit.

## 1 Introduction

The implications of COVID-19 extend far beyond its immediate physical health effects. Uncertainty and fear surrounding the disease and its effects, in addition to a lack of consistent and reliable information, contribute to rising levels of anxiety and stress (Torales et al., 2020). Policies designed to help contain the disease also have significant consequences. Social distancing policies and lockdowns lead to increased feelings of isolation and uncertainty (Huremović, 2019). They have also triggered an economic downturn (Şahin et al., 2020), resulting in soaring unemployment rates and causing many to experience financial stress. Therefore, in addition to the profound effects on physical health around the world, psychiatrists have warned that we should also brace for a mental health crisis as a result of the pandemic (Qiu et al., 2020; Greenberg et al., 2020; Yao et al., 2020; Torales et al., 2020).

---

[*]Denotes equal contribution.

Indeed, the literature on the impact of past epidemics indicates that they are associated with a myriad of adverse mental health effects. In a review of studies on the 2002-2003 SARS outbreak, the 2009 H1N1 influenza outbreak, and the 2018 Ebola outbreak, Chew et al. (2020) found that anxiety, fear, depression, anger, guilt, grief, and post-traumatic stress were all commonly observed psychological responses. Furthermore, many of the factors commonly cited for inducing these responses are applicable to the COVID-19 setting. These include: fear of contracting the disease, a disruption in daily routines, isolation related to being quarantined, and uncertainty regarding the disease treatment process and outcomes, the well-being of loved ones, and one's economic situation.

While disease outbreaks pose a risk to the mental health of the general population, research suggests that this risk is heightened for those with pre-existing mental health concerns. People with mental health disorders are particularly susceptible to experiencing negative mental health consequences during times of social isolation (Usher et al., 2020). Further, as Yao et al. (2020) warn, they are likely to have a stronger emotional response to the feelings of fear, anxiety, and depression that come along with COVID-19 than the general population.

Given the potential for the COVID-19 outbreak to have devastating consequences for mental health, it is critical that we work to understand its psychological effects. In this work, we use Reddit, a popular social media platform, to study how COVID-19 has impacted the behavior of groups of users who express mental health concerns. We analyze the content of discussions (COVID-related discussions, psycholinguistic categories, and topics) as well as the volume of communication (daily user count) and find notable changes in each category. Some of these changes appear in multiple mental health subreddits, but some are more specific to individual

communities that relate to specific diagnoses. We believe that our findings can help us better understand and potentially alleviate the negative mental health effects of the pandemic; for instance, this type of analysis could help moderators to more effectively support users through future crises. To the best of our knowledge, the method that we propose has not been used previously to study changes in mental health subreddits, and could be applied to understand the effects of other major events like political elections and natural disasters.

## 2 Related Work

### 2.1 Linguistic Analysis and Mental Health

There is a considerable body of research that examines the relationship between language use and mental health, including work dating back several decades. For example, Bucci and Freedman (1981) and Weintraub (1981) observed an increased usage of first person singular pronouns in individuals with depression. Oxman et al. (1982) showed that they could distinguish between paranoia and depression by applying linguistic analysis to speech.

Since then, advances in tools for text analytics have led to increased research in this area. Notably, the Linguistic Inquiry and Word Count (LIWC) (Pennebaker et al., 2001) is a widely used computerized text analysis tool that has been validated for psycholinguistic analysis. Some of the earliest studies using LIWC analyzed written text. For instance, researchers have used LIWC to study linguistic patterns in essays written by college students with and without depression (Rude et al., 2004) or in poems written by suicidal vs non-suicidal poets (Stirman and Pennebaker, 2001). More recently, there has been a proliferation of studies applying LIWC to online text, including social media data. LIWC has been used to study language patterns on social media for a variety of mental health disorders, including depression, anxiety, suicidality, and bipolar disorder (De Choudhury et al., 2013; Shen and Rudzicz, 2017; Coppersmith et al., 2014, 2016). In addition to LIWC, other methods used to study the linguistic patterns of mental illness include character and word models (Coppersmith et al., 2014; Tsugawa et al., 2013) and topic modeling (Resnik et al., 2015; Preoţiuc-Pietro et al., 2015).

### 2.2 Studying Mental Health via Social Media

In the past decade, social media has emerged as a powerful tool for understanding human behav-

ior, and correspondingly mental health. A growing number of studies have applied computational methods to data collected from social media platforms in order to characterize behavior associated with mental health illnesses and to detect and forecast mental health outcomes (see Chancellor and De Choudhury (2020) for a comprehensive review).

Reddit is a particularly well-suited platform for studying mental health due to its semi-anonymous nature, which encourages user honesty and reduces inhibitions associated with self-disclosure (De Choudhury and De, 2014). Additionally, Reddit contains subreddits that act as mental health support forums (e.g., r/Anxiety, r/depression, r/SuicideWatch), which enable a more targeted analysis of users experiencing different mental health conditions. A number of existing works have focused on characterizing patterns of discourse within these mental health communities on Reddit. These include studies that have analyzed longitudinal trends in topic usage and word choice (Chakravorti et al., 2018), the relationship between user participation styles and topic usage (Feldhege et al., 2020), and the discourse patterns specific to self-disclosure, social support, and anonymous posting (Pavalanathan and De Choudhury, 2015; De Choudhury and De, 2014).

Other studies of Reddit mental health communities have aimed to quantify and forecast changes in user behavior. De Choudhury et al. (2016) presented a model for predicting the likelihood that users transition from discussing mental health generally to engaging in suicidal ideation. Li et al. (2018) analyzed linguistic style measures associated with increasing vs decreasing participation in mental health subreddits over the course of a year. Kumar et al. (2015) examined how posting activity in r/SuicideWatch changes following a celebrity suicide. Our work similarly focuses on analyzing temporal patterns in user activity, but we aim to characterize changes associated with COVID-19.

### 2.3 Mental Health and COVID-19

Since the first cases of COVID-19 were reported in December 2019, there have been a number of preliminary studies of its impact on mental health. In a survey of the general public of China, a majority of respondents perceived the psychological impact of the outbreak to be moderate-to-severe and about one-third reported experiencing moderate-to-severe anxiety (Wang et al., 2020). Studies of the

impact of COVID-19 among residents of Liaoning Province, China (Zhang and Ma, 2020) and the adult Indian population (Roy et al., 2020) also found notable rates of mental distress.

There is a set of studies that have examined the mental health consequences of COVID-19 by analyzing online behaviors. Jacobson et al. (2020) explored the short-term impact of stay-at-home orders in the United States by analyzing changes in the rates of mental health-related Google search queries immediately after orders were issued. Their results showed that rates of mental health queries increased leading up to the issuance of stay-at-home-orders, but then plateaued after they went into effect; however they did not consider the longer-term implications of the stay-at-home orders on mental health. Li et al. (2020) measured psycholinguistic attributes of posts on Weibo, a Chinese social media platform, before and after the Chinese National Health Commission declared COVID-19 to be an epidemic. Their findings showed that expressions of negative emotions and sensitivity to social risks increased following the declaration. Wolohan (2020) used a Long Short-Term Memory model to classify depression among Reddit users in April 2020, finding a higher than normal depression rate.

Our work similarly aims to measure changes in online behavior as a means of understanding the relationship between COVID-19 and mental health. However, two notable differences are: (1) instead of analyzing the short-term impact of a specific COVID-related event, we examine more general changes that have occurred during a three-month period of the outbreak; and (2) we focus our analysis on activity within mental health forums, which allows us to examine the impact of COVID-19 specifically on individuals who have expressed mental health concerns.

## 3 Data

We collect Reddit posts from three mental health subreddits using the Pushshift API[1] (Baumgartner et al., 2020): r/Anxiety, r/depression, and r/SuicideWatch, from January 2017 to May 2020. The reasons for analyzing these three subreddits are twofold: first, over the three and a half years represented in our data, these subreddits have a significant amount of activity ($\geq 40$ posts every

|      | r/Anxiety | r/depression | r/SuicideWatch |
|------|-----------|--------------|----------------|
| 2017 | 95        | 279          | 91             |
| 2018 | 164       | 449          | 188            |
| 2019 | 211       | 622          | 285            |
| 2020 | 243       | 618          | 370            |

Table 1: Average number of posts per day across the three subreddits in our dataset.

day), making it feasible to treat daily values as a time series. Second, because the subreddits provide support for different mental health disorders, their users may have been affected differently by COVID-19. We separate the data into two time periods: pre-COVID (January 1, 2017 - February 29, 2020) and post-COVID (March 1, 2020 - May 31, 2020), roughly delineating when COVID-19 began to have a serious impact on those in the United States, where the majority of Reddit users are concentrated.[2] This choice of dates was informed by our analysis of the rates at which COVID-19 related words were discussed in each subreddit (see Section 5.1), which we found hovered around 0-5% before rising sharply near the beginning of March.

We exclude posts where the author or text is marked as '[removed]' or '[deleted]', because posts with deleted authors offer no value for user count metrics, and deleted content means that we are unable to capture linguistic signals (see Section 4.1 for more details on these metrics). Figure 1 shows the average number of daily posts for r/Anxiety, r/depression, and r/SuicideWatch.

## 4 Methodology

Our goal is to identify how mental health subreddit activity has changed during the pandemic. We first create time series for a number of metrics that could be affected by the pandemic, encompassing activity levels and text content (Section 4.1). We then use a time series intervention analysis technique to determine whether there are significant changes in our metrics during the pandemic (Section 4.2).

### 4.1 Reddit Activity Metrics

We begin by creating a lexicon of words that are commonly used to refer to COVID-19. This allows us to determine the extent to which users in each subreddit are discussing COVID-19, and also gives us a clearer idea of when COVID-19 began

---

[1]As with other social media datasets, there may be noise in the form of API changes and data removed after collection. For the dates involved in our study, static Pushshift dump files were not yet available.

[2]https://www.alexa.com/siteinfo/reddit.com

to directly affect discussion in the mental health subreddits. We based the lexicon on a set of twitter search keywords from Huang et al. (2020), and added six additional words that we believed would be indicative of discussion about COVID-19 (see the full lexicon in Appendix A).

To study changes in the number of users seeking mental health support in subreddits, we record the author usernames for each post in our dataset. Since individuals can create multiple accounts under different usernames, the number of unique usernames associated with posts is likely not equal to the true number of unique users; however, it is a reasonable proxy.

To study changes in content that occur during the pandemic, we use the LIWC lexicon (Pennebaker et al., 2015) and Latent Dirichlet Allocation (LDA) topic modeling (Blei et al., 2003). The LIWC lexicon consists of seventy-three hierarchical psycholinguistic word categories, encapsulating properties including linguistic categories (e.g., 1st person plural pronouns, verbs), emotions (e.g., anxiety, sadness), time (e.g., present, future), and personal concerns (e.g., work, money, death). To capture the discussion topics that are common in the r/Anxiety, r/depression, and r/SuicideWatch subreddits specifically, we train a topic model on posts from these subreddits. We ensure that discussions from each of the subreddits are equally represented in our training dataset by randomly downsampling the posts from the subreddits with more data. We use the implementation of LDA topic modeling provided in the MALLET toolkit (McCallum, 2002) and train models with $k = 5, 10, .., 40$ topics. We select a single model to use in our analysis by examining their coherence scores, a measure of the semantic similarity of high probability words within each topic (Mimno et al., 2011). As coherence scores tend to increase with increasing $k$, we select $k$ as the first local maxima of coherence scores, which we found to be $k = 25$.

In Appendix B, we show the 25 topics obtained from our topic model, along with the highest probability words associated with each topic. We also provide labels that summarize the essence of each topic, which we created by examining their representative words. Common themes of discussion include: daily life concerns (e.g., school, work, sleep and routine), personal relationships (e.g., friends, family, relationships), and mental health struggles (e.g., anxiety, suicide, medical treatment).

When using text from posts, we remove special characters and sequences, such as newlines, quotes, emails, and tables. To represent the text of a post, we concatenate the title with the text content, as was done in prior work (Chakravorti et al., 2018). We apply additional pre-processing steps for our topic modeling analysis: (1) we remove a set of common stopwords that do not appear in the LIWC lexicon (we kept those in LIWC as they have been found to have psychological meaning), (2) we form bigrams from pairs of words that commonly appear together, and (3) we lemmatize each word.

## 4.2 Time Series Analysis

We treat the task of identifying changes in subreddit activity patterns as a time series intervention analysis problem. Our basic approach involves: (1) fitting a time series model to the pre-COVID observations for each of the metrics described above and then (2) examining how the values forecasted by the model compare to the observed values during the post-COVID time period. It is worth noting that the one study we found examining the impact of an event on activity within mental health subreddits employs a different approach: they use a t-test to compare the observations from "before" vs "after" the event (Kumar et al., 2015). However, their problem setup differs from ours in that they consider a much shorter period of time (four weeks total), so the effects of seasonality (regular changes that recur each year) and longer-term trends are likely reduced. In contrast, we find that there is often a strong trend over time and seasonal component in our data, making a direct comparison of two time periods with a t-test unreliable.

We smooth each time series and remove day-of-week related fluctuations by computing a seven-day rolling mean over the time series. We use the Prophet model (Taylor and Letham, 2018) to create a model of the period before COVID-19. This model was initially created by Facebook to forecast time series on their platform, such as the number of events created per day or the number of active users; we find that our time series, also compiled from social media, have many similar properties. The Prophet model is an additive regression model with three components:

$$y(t) = g(t) + s(t) + h(t) + \epsilon_t \qquad (1)$$

The trend is encapsulated by $g(t)$, a piecewise linear model. The seasonality of the data is cap-

tured by $s(t)$, which is approximated using a Fourier series. As we smooth our data on a weekly basis, we utilize only yearly seasonality, excluding the optional weekly and daily seasonality components. The third term, $h(t)$, represents holidays; we find that adding the default list of US holidays provided by Prophet reduces error for most our our time series in the pre-COVID period, likely because the Reddit population is centered in the United States. Finally, $\epsilon_t$ represents the error, in this case fluctuations in the time series that are not captured by the model.

After training the model on the pre-COVID data, we predict values for the post-COVID period. If we assume that there is no change during this time period, we would expect the predicted values to be near the true values, given that the model does a good job fitting the trend and seasonal components. The model computes uncertainty intervals over the predicted values by simulating ways in which the trend may change during the period of the forecast. We use this method to compute the 95% prediction interval. Our null hypothesis is that there has been no change in trend. In this case, we would expect 5% of the data in the post-COVID period to fall outside of the prediction interval. Our alternate hypothesis is that there was a change in the trend of the time series (which may be attributable to COVID-19). In this case, more than 5% of the data in the post-COVID period will fall outside of the prediction interval. We apply a one-sample proportion test to assess whether the proportion of observations outside of the prediction interval in the post-COVID period is significantly greater than 5%. The details of this test are in Appendix C.

# 5 Results and Discussion

## 5.1 How often do people in different mental health subreddits discuss COVID-19?

Using our COVID-19 lexicon (Section 4.1), we compute the percentage of posts per day that mention any words related to COVID-19, as shown in Figure 1. We see that COVID-19 began to have a serious impact on discussions in all three subreddits around the beginning of March 2020, as is clear from the spikes in Figure 1. Although COVID-19 is discussed on all subreddits, we see a stark difference in the volume of discussion across each of them; in r/Anxiety, discussion of COVID-19 is more frequent than it is in r/depression or r/SuicideWatch, and begins earlier.

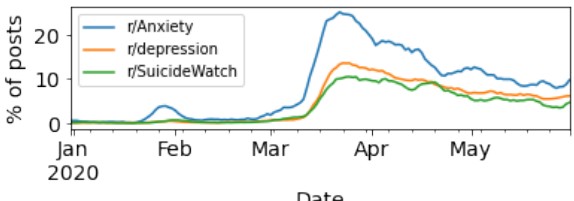

Figure 1: Percent of posts mentioning COVID-19 related words across mental health subreddits.

**Discussion** When choosing the date to consider as the beginning of the post-COVID period in our time series analysis, we considered March 1st, 2020 as a sensible date, as it aligns with the time at which the United States (where the majority of Reddit users reside) began to take COVID-19 seriously. March 1st closely followed the first announced COVID-19 death in the United States on February 28th, 2020, and preceded state lockdowns and school closures. The spikes at the beginning of March suggest that this date also reflects the time at which COVID-19 began to have a notable impact on mental health subreddit discussions.

Although most COVID-19 related discussion started in March, we also see that a small spike in discussion rates occurred earlier in r/Anxiety. This suggests that users in this subreddit began to notice some impact from COVID-19 in late January, when reports of lockdowns in China first appeared in the news. Based on the early start and elevated rate of COVID-19 discussion within r/Anxiety, we conclude that all of our metrics are likely to be more strongly affected by COVID-19 in r/Anxiety.

## 5.2 Has COVID-19 changed the number of users seeking support in mental health subreddits?

We report the daily number of unique users who posted in each subreddit in Figure 2. We observe an increase in the number of users who posted in the r/Anxiety subreddit in the post-COVID period. Meanwhile, in both r/depression and r/SuicideWatch, we find significant decreases in the number of users who posted. In r/depression, we observe a substantial drop in posting rates around mid-March. Activity in this subreddit remains abnormally low into late-April, when it starts to revert back towards the forecasted values. In r/SuicideWatch, the drop in user activity is less extreme, and we see that the activity levels eventually return to their predicted values.

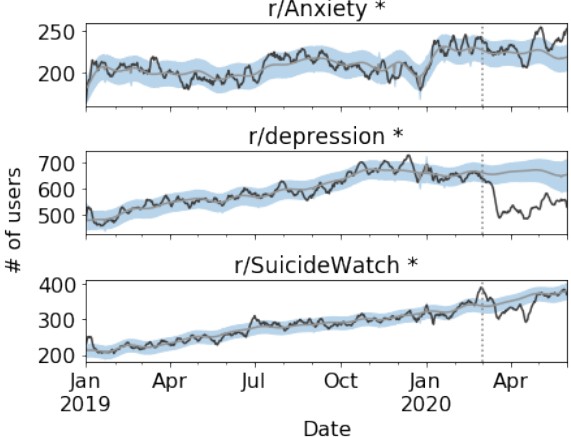

r/Anxiety *

r/depression *

r/SuicideWatch *

# of users

Jan 2019 — Apr — Jul — Oct — Jan 2020 — Apr

Date

Figure 2: Daily active users over time. The grey line is the Prophet forecast, the shaded area is the 95% prediction interval, and the black line is the true value. Subreddits marked with * have a statistically significant percentage of outliers ($\alpha = 0.05$).

| r/Anxiety | | | r/depression | | | r/SuicideWatch | | |
|---|---|---|---|---|---|---|---|---|
| Category | % Outliers | | Category | % Outliers | | Category | % Outliers | |
| Motion* | 79 | ↓ | You* | 55 | ↓ | Prep* | 33 | ↑ |
| Work* | 73 | ↓ | Conj* | 51 | ↓ | Space* | 33 | ↑ |
| I* | 68 | ↓ | Motion* | 45 | ↓ | Netspeak* | 23 | ↑ |
| Body* | 61 | ↑ | Quant* | 43 | ↑ | Assent* | 23 | ↓ |
| PPron* | 54 | ↓ | Family* | 40 | ↑ | Informal* | 22 | ↑ |
| Relativ* | 54 | ↓ | Article* | 39 | ↓ | Cause* | 20 | ↑ |
| We* | 50 | ↑ | Pronoun* | 38 | ↑ | Affiliation | 17 | ↓ |
| Bio* | 49 | ↑ | Reward* | 36 | ↓ | FocusFuture | 16 | ↓ |
| Percept* | 42 | ↑ | Feel* | 35 | ↓ | NegEmo | 15 | ↑ |
| Certain* | 41 | ↑ | FocusPast* | 33 | ↑ | Conj | 15 | ↓ |

Table 2: Ten LIWC categories with the highest proportion of outliers in each subreddit. Arrows mark the direction in which the mean of the outliers shifted from the predicted mean. Categories marked with * have a statistically significant percentage of outliers (with Bonferroni correction; $\alpha = 0.05$ before correction).

**Discussion** The increase in users posting on r/Anxiety is consistent with prior work that has found that epidemics often lead to increased rates of anxiety (Torales et al., 2020). One explanation for the reduction of activity within r/depression could be that fewer users are depressed and don't feel the need to post on the support forum. If this is the case, our findings contrast with prior work that found that depressive symptoms are commonly observed during pandemics (Chew et al., 2020). However, there are multiple possible alternatives; for example, depression can also cause people to socially withdraw (Mayo Clinic, 2018), so an increase in depression rates could lead to a reduction in posting activity. Another finding from prior work is that *delayed* depression is common following disaster events (Pennebaker and Harber, 1993; Nandi et al., 2009). Our analysis covers only the beginning of the pandemic, so it likely wouldn't capture this phenomenon. Additional analysis focused on the causes driving the reduction in activity and how this pattern changes in the long-term is needed to make a more conclusive statement about the effects of COVID-19 on depression.

## 5.3 Has COVID-19 led to changes in the discussions users have surrounding mental health?

To determine what changes have occurred in conversations surrounding mental health, we use two types of features: LIWC categories and topics obtained from an LDA model. The LIWC features give us a better idea of how common language dimensions have changed, while the LDA-derived topics allow us to explore areas of discussion that are typically of concern in these subreddits. LIWC has been used extensively in mental health analysis, and there are some LIWC categories and LDA-derived topics that overlap, such as ANXIETY, DEATH, and FAMILY, but there are also unique categories covered by each method, such as WE and MOTIVATION. For each of the metrics, we examine changes that have occurred since COVID-19 by computing the proportion of outliers produced by our forecasting model (see Section 4.2) in the post-COVID period. We acknowledge that this analysis may occasionally capture misleading changes. For example, the death keyword may yield changes in the suicide topic (see Appendix B) that are actually related to infectious disease, and observing increased mentions of family does not indicate the polarity of their sentiment. We leave it to future work to do a more in-depth analysis of the context surrounding specific outliers that are detected.

### 5.3.1 LIWC Analysis

In Table 2, we show the ten LIWC categories with the most outliers (outside of the 95% prediction interval) from March to May of 2020. We observe a lack of consistency between the subreddits, both in the number and direction of the outliers.

We see decreases in r/Anxiety and r/depression in the MOTION category. Categories such as BIO and BODY tend to increase in r/Anxiety; however, this pattern is not present in other subreddits. We see consistent changes in time orientation (e.g., FOCUSPAST, FOCUSFUTURE) across subreddits; a higher focus on the past in r/depression, and a

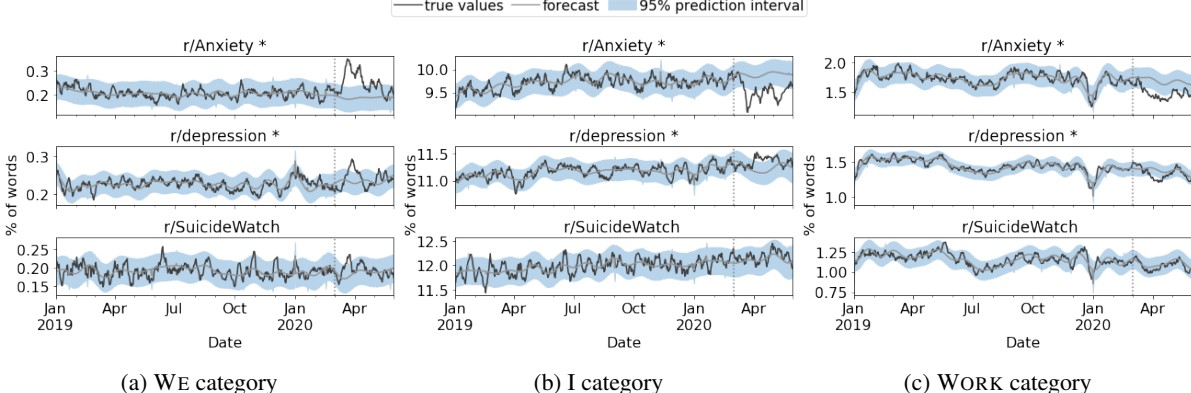

Figure 3: Average daily percent of words across posts from a selection LIWC categories over time. The grey line is the Prophet forecast, the shaded area is the 95% prediction interval, and the black line is the true value. Subreddits marked with * have a statistically significant percentage of outliers (with Bonferroni correction; $\alpha = 0.05$ before correction).

lower focus on the future in r/SuicideWatch. While it is not among the categories with the most outliers, there is a statistically significant drop in FO­CUSFUTURE on r/Anxiety and r/depression. We also see changes in pronoun usage; the most no­table and consistent change across the subreddits is that the usage of WE increases significantly, espe­cially in the early period of COVID-19 (Figure 3a). While there is a significant decrease in I words in r/Anxiety, there is in fact an increase in r/depression (Figure 3b). Finally, we see a notable drop in the WORK category (Figure 3c).

**Discussion** We see changes in some categories that appear to be directly related to the new expe­rience of living during a global pandemic under social distancing rules; this includes the decrease in MOTION, which makes sense as people are trav­eling and moving around far less. The increase in categories such as BIO and BODY within r/Anxiety may reflect concerns regarding the physical health implications of COVID-19. Moreover, it appears that physical health concerns are especially salient for people who experience anxiety, as the rise in these categories is not present in the other subred­dits. The statistically significant drop in FOCUSFU­TURE within r/Anxiety and r/depression indicates that users are less inclined to speak about their con­cerns for the future in light of the more pressing current concerns related to the pandemic.

The sharp increase in WE (Figure 3a) indicates a general feeling of community and togetherness, which speaks positively to the support that those in these mental health communities are getting during the pandemic. This finding aligns with a study by Zhang and Ma (2020) on the effects of COVID-19 on mental well-being in China, which found that participants received increased support from friends and family during the pandemic. In addi­tion, seeking social support was listed as a common coping strategy during infections disease outbreaks by Chew et al. (2020). An increase in "we" is not specific to mental health communities; researchers have found increases in usage of the pronoun dur­ing the early stages of COVID-19 on other sub­reddits (Ashokkumar and Pennebaker, 2020). The decrease in I words in r/Anxiety is accompanied by an increase in r/depression (Figure 3b). The increase of usage of the I pronoun is concerning because it has been shown to correlate with depres­sion, indicating that an increase in its use could be related to worsening symptoms (Rude et al., 2004).

The drop in discussion of WORK (Figure 3c) is unexpected, as the economic downturn could be a significant motivator of posts. The drop indicates that up to this point, the stress and change associ­ated with adapting to working from home, or worse, losing one's job has not been a frequent topic of discussion in these forums. This drop could be due to a decrease in work-related stressors, which have been shown to cause anxiety and depression (Melchior et al., 2007; Cherry, 1978), or it could simply indicate that the stressors became secondary to other concerns. It is also possible that compared to the general population, Reddit users are more likely to have jobs that can be done remotely dur­ing the pandemic, as they are more likely to have

college degrees than the general population.[3]

### 5.3.2 Topic Analysis

We report the ten topics with the highest proportion of outliers for each subreddit during the COVID-19 period (March to May 2020) in Table 3. One notable trend is an increase in the amount of discussion related to family; we find that the FAMILY AND HOME topic increased significantly in all three subreddits and the FAMILY AND CHILDREN topic increased significantly in r/depression. Figure 4b shows how the usage of the FAMILY AND HOME topic has changed since January 2019 within each subreddit. While there are noticeable increases in all three subreddits, we a see particularly large spike in r/Anxiety starting around mid-March. Within all subreddits, we see a significant decrease in the TRANSPORT AND DAILY LIFE topic (see Figure 4c), which is associated with words such "drive," "car," "time," and "day". Mirroring the reduction of WORK-related language we observed in Section 5.3.1, we also find that there has been a significant decrease in discussion of the SCHOOL and WORK topics within the r/Anxiety and r/depression subreddits.

We observe significant changes in topics that are explicitly related to mental health. One of the most prominent trends is a significant increase in discussions of ANXIETY and its symptoms (keywords include: "panic," "heart," and "chest"). As seen in Figure 4a, we see a spike in ANXIETY in mid-March in all three subreddits; however, whereas we see a return to a typical level in both r/depression and r/SuicideWatch, within the r/Anxiety, ANXIETY discussion rates have remained abnormally high all the way through the end of May. We find that both INFORMATION SHARING (keywords include: "post," "read," "share," "find," and "hope") and COMMUNICATION (keywords include: "talk," "call," and "message") have become more frequent topics of discussion.

**Discussion** Several of the results in Table 3 seem to reflect the disruption to normal daily life caused by COVID-19 and the resulting quarantine measures. This includes the increase in the FAMILY AND CHILDREN topic (Figure 4b), which is largely expected, as quarantine policies implemented to help contain COVID-19 have resulted in many people spending more time at home and with family

than they previously had. While there are noticeable increases in all three subreddits, we see a particularly large spike in r/Anxiety starting around mid-March. Prior studies on disease outbreaks have found that uncertainty regarding the well-being of loved ones is a common source of anxiety during epidemics, which may help to explain this finding (Chew et al., 2020). Another contributing factor may be the emergence of new family responsibilities, such as childcare and home-schooling, that many people have had to take on in the face of closures caused by the pandemic. The decrease in the TRANSPORT AND DAILY LIFE topic (Figure 4c) is intuitive; quarantine practices following COVID-19 have led to a large reduction in driving and other forms of transportation (Domonoske and Adeline, 2020) and, more generally, to a disruption in daily lifestyles. To the extent that these results indicate an abandonment of routine, they are somewhat concerning, as evidence from prior outbreaks suggests that getting back into normal routines helps to reduce loneliness and anxiety during quarantines (Huremović, 2019). The decreases in discussion of the SCHOOL and WORK topics may indicate that these previously common sources of stress have now become secondary concerns compared to the more immediate concerns associated with COVID-19.

The increases in the ANXITETY topic, especially on r/Anxiety, are aligned with existing research that has found that anxiety and the somatic symptoms associated with it are common psychological responses to epidemics (Chew et al., 2020). Further, studies of prior epidemics have found that feelings of anxiety and fear can persist even after the disease itself has been contained (Usher et al., 2020).

The increase in the INFORMATION SHARING and COMMUNICATION topics may be tied to the effects of social distancing measures, which have limited in-person interactions and led people to increasingly turn to digital methods of communication. These observations may also reflect a desire to seek out information related to COVID-19; individuals who experience health anxiety are more likely to exhibit online health information seeking behavior (McMullan et al., 2019). The increase in mentions of words related to social media (e.g., "post," "share") is somewhat worrisome; studies of disaster events have found that both more frequent social media use and exposure to conflicting information online (a widely acknowledged issue with COVID-

---

[3]https://www.statista.com/statistics/517222/reddit-user-distribution-usa-education/

| r/Anxiety | | | r/depression | | | r/SuicideWatch | | |
|---|---|---|---|---|---|---|---|---|
| Topic | % Outliers | | Topic | % Outliers | | Topic | % Outliers | |
| Transport and Daily Life* | 88 | ↓ | Family and Home* | 83 | ↑ | Transport and Daily Life* | 70 | ↓ |
| Anxiety* | 75 | ↑ | Transport and Daily Life* | 82 | ↓ | Family and Home* | 50 | ↑ |
| Information Sharing* | 68 | ↑ | Information Sharing* | 62 | ↑ | Friends* | 32 | ↓ |
| School* | 62 | ↓ | Work* | 55 | ↓ | Anxiety* | 28 | ↑ |
| Family and Home* | 48 | ↑ | Suicide* | 50 | ↓ | Family and Children | 22 | ↑ |
| Life and Philosophy* | 34 | ↑ | Sleep and Routine* | 50 | ↓ | "Game-over" Mentality and Swearing | 15 | ↑ |
| Work* | 33 | ↓ | Communication* | 48 | ↑ | Suicide | 14 | ↓ |
| "Game-over" Mentality and Swearing* | 29 | ↓ | "Game-over" Mentality and Swearing* | 39 | ↑ | Worry | 14 | ↑ |
| Experience and Mental State* | 27 | ↑ | Medical Treatment* | 36 | ↓ | Communication | 13 | ↑ |
| Motivation* | 22 | ↓ | Family and Children* | 34 | ↑ | People and Behavior | 13 | ↑ |

Table 3: Ten topics with the most outliers for r/Anxiety, r/depression, and r/SuicideWatch. Arrows mark the direction in which the mean of the outliers shifted from the predicted mean. Topics marked with * have a statistically significant percentage of outliers (with Bonferroni correction; $\alpha = 0.05$ before correction).

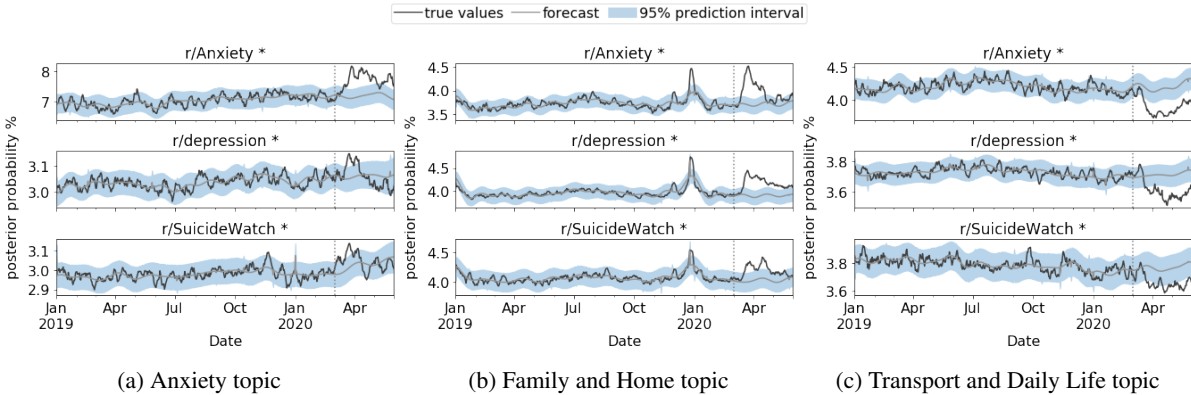

(a) Anxiety topic  (b) Family and Home topic  (c) Transport and Daily Life topic

Figure 4: Average daily posterior probability of selected topics in posts over time. The grey line is the Prophet forecast, the shaded area is the 95% prediction interval, and the black line is the true value. Subreddits marked with * have a statistically significant percentage of outliers (with Bonferroni correction; $\alpha = 0.05$ before correction).

19 (Kouzy et al., 2020)) lead to higher stress levels (Torales et al., 2020). However, the rise of the INFORMATION SHARING topic, especially in it's relation to words like "share," "hope," and "story," could also be indicative of a collective coping process, in which individuals come together for social support. As noted in Section 5.3.1, this type of coping strategy has frequently been observed during past disease outbreaks (Chew et al., 2020) and may also be reflected by the increase in the usage of WE we saw for discussions in r/Anxiety.

## 6 Conclusions

In this study, we examined how COVID-19 has influenced the online behavior of individuals who discuss mental health concerns by analyzing activity within the r/Anxiety, r/depression, and r/SuicideWatch communities on Reddit. We found substantial evidence of increases in anxiety; we observed an increase in user activity in r/Anxiety, as well as significant increases in discussions of anxiety and the symptoms associated with it. Interestingly, we observed a decrease in activity within the

r/depression and r/SuicideWatch subreddits. The literature on the impact of disease outbreaks on depression rates contains somewhat contradictory findings; we therefore believe that this is an interesting area for future work.

We also observed interesting changes in the content of discussions within each subreddit. Our results suggest that concerns related to COVID-19, such as health and family, have become more prominent discussion topics compared to other common concerns, such as work and school, which have generated relatively less discussion since the outbreak. While our findings largely confirm the warnings offered by psychiatrists regarding the potential for COVID-19 to have an adverse effect on mental health, we also found some reason for optimism; increases in the usage of WE as well as the INFORMATION SHARING topic (associated with words such as "story" and "hope"), suggest a heightened sense of community and shared experience, which may help individuals cope with these stressful times.

## Acknowledgment

We are grateful to the Michigan AI lab for discussions that led to this project, and to the statistical consultants at CSCAR who helped with developing our process for hypothesis testing, especially Kerby Shedden and Thomas Fiore. This material is based in part on work supported by the Precision Health initiative at the University of Michigan, the NSF (grant #1815291), and the John Templeton Foundation (grant #61156). Any opinions, findings, conclusions, or recommendations in this material are those of the authors and do not necessarily reflect the views of the Precision Health initiative, the NSF, or the John Templeton Foundation.

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

## A   COVID-19 Lexicon

The following terms from Huang et al. (2020) are included in our COVID-19 lexicon: 2019-ncov, 2019ncov, coronavirus, COVID, COVID-19, COVID19, mers, sars, SARS2, SARSCOV19, wuflu, Wuhan. We add the following terms: corona, outbreak, pandemic, rona, sars-cov-2, virus. We ignore case when counting occurrences, and therefore exclude duplicate terms that only differ in their case.

# B Topics identified by LDA Model

Figure 4 shows the topics identified by the LDA
model.

| Topic Label | High Probability Words |
|---|---|
| School | school, year, college, high, class, fail, parent, study, grade, start |
| Relationships | love, relationship, girl, guy, good, girlfriend, break, date, meet, find |
| Experience and Mental State | experience, situation, mind, part, brain, lead, state, feeling, sense, learn |
| Communication | talk, call, time, phone, send, text, give, back, speak, message |
| People and Behavior | people, make, person, care, thing, understand, problem, wrong, act, attention |
| Feelings | happy, tired, cry, anymore, sad, make, hurt, depressed, stop, feeling |
| "Game-over" Mentality and Swearing | hate, fuck, shit, fucking, die, wanna, stupid, kill, literally, idk |
| Transport and Daily Life | drive, time, back, car, drink, start, walk, home, run, day |
| Time | year, month, start, time, back, ago, day, week, past, couple |
| Worry | thought, mind, fear, worry, head, afraid, scared, scare, stop, happen |
| Friends | friend, talk, people, good, social, play, make, close, hang, group |
| Anxiety | anxiety, attack, panic, anxious, heart, symptom, calm, chest, experience, stress |
| Medical Treatment | anxiety, medication, doctor, therapy, med, therapist, experience, week, mg, work |
| Body and Food | eat, body, eye, face, hand, head, sit, food, weight, walk |
| - | bad, thing, make, time, lot, happen, pretty, good, stuff, kind |
| Life and Philosophy | life, world, hope, exist, dream, human, live, pain, love, real |
| Depression and Mental Illness | depression, mental, issue, health, problem, struggle, deal, bad, suffer, year |
| Life Purpose | life, live, end, anymore, point, family, reason, care, worth, future |
| Motivation | thing, time, make, good, find, hard, work, enjoy, change, motivation |
| Work | work, job, money, pay, quit, find, afford, interview, company, month |
| Family and Children | year, family, mother, kid, parent, child, life, father, young, age |
| Information Sharing | post, read, write, find, hope, give, share, story, reddit, long |
| Family and Home | leave, mom, home, move, house, dad, family, live, parent, stay |
| Sleep and Routine | day, sleep, night, hour, wake, today, bed, morning, work, week |
| Suicide | kill, die, suicide, pain, suicidal, end, attempt, cut, plan, dead |

Table 4: Topics identified by the LDA topic model. For each topic, we provide a summary label and the ten most probable words. We omit labels for topics whose keywords did not have a clear interpretation.

## C   Statistical Significance Test

We apply a one-sample proportion test to assess whether the proportion of observations outside of the prediction interval in the post-COVID period is significantly greater than 5%. This test assumes that the observations are independent; however, we find that there is order-1 autocorrelation in our data. We therefore apply a correction for order-1 autocorrelation (Zwiers and Von Storch, 1995) when computing the z-test statistic. The corrected test statistic is:

$$z = \frac{\hat{p} - p_0}{\sqrt{p_0(1 - p_0)(1 + r)/n(1 - r)}} \quad (2)$$

where $\hat{p}$ is the proportion of observations outside of the prediction interval in the post-COVID period, $p_0 = 0.05$, $n$ is the number of observations in the post-COVID period, and $r$ is the lag-1 correlation coefficient of the pre-COVID data.

We use a Bonferroni correction when determining statistical significance for our discussion content metrics, as we ran almost 300 tests. $M = 294$, which is the number of LIWC categories and topics, multiplied by the number of subreddits. Our corrected $\alpha = 0.05/294 = 1.7 \times 10^{-5}$.