# OpenReview forum: "Quantifying the Effects of COVID-19 on Mental Health Support Forums"
_EMNLP/2020/Workshop/NLP-COVID — NLP-COVID19-EMNLP Oral_

### Official Review · AnonReviewer2 · 2020-09-12
**Clear contribution with opportunities for improvement**

**Rating:** 8
**Confidence:** 4

**Review:**

#### Recommendation
Paper has room for improvement, but makes clear additions to the field.

#### Summary
**Strengths**
1. Comparison between anxiety, depression, suicide is very relevant
2. Methods are well described
3. Use of LIWC to supplement topic-modeling was a good decision
**Weaknesses**
1. Traditional **Discussion** section is intermixed with the section 5: **Findings**
2. An embedded topic model (Dieng, Ruiz, and Blei 2019) offers benefits over LDA
3. Discussion of LIWC and depression/anxiety prediction for NLP is not particularly thorough

#### Review
The authors examine how the content of three subreddits related to mental-health help seeking changes as a result of the COVID-19 pandemic. To do so, they combine established methods for topic analysis (LDA) and psychological assessment of text (LIWC) with a time-series prediction approach (Prophet). Their analysis contributes to the field and their approach is satisfactorily documented and executed; however, the authors could improve their work with better organization and use of more current methods.

The authors comparison of anxiety and depression is an important contribution. USC's Understanding America Survey COVID-19 is doing this as well. Useful to show that NLP can be used to the same end. The authors longitudinal analysis of established NLP techniques, LIWC and LDA, usefully allows them to identify topic-changes across the various subreddits.

The paper could be made more accessible through a more typical distinction between analysis and discussion. As it is written, the authors analysis and discuss implications simultaneously--leading to temporary uncertainty over what is an evidence-backed finding and what is author speculation.

The authors could also update their methodological choices, e.g., using [Embedding Topic Modeling](https://arxiv.org/pdf/1907.04907.pdf)(Dieng, Ruiz, and Blei 2019) in place of LDA. fastText embeddings would be a natural choice for this, as the authors have already cited Wolohan, 2020.

Lastly, the authors could also discuss more thoroughly the history of LIWC and word-count approaches in NLP for depression / mental health detection. Rude et al. (2004) is cited but only as a possible explanation for a finding. That paper is a significant forebearer to this work.

#### Reproducability
This paper is highly reproducible, though associated code or Reddit post-IDs would help.

---

> ### Author Response · Authors · 2020-09-28
> **Response to Reviewer 2**
>
> We are hesitant to alter section 5 to the extent of creating separate results and discussion sections because we believe that being able to discuss the results and implications together allows the writing to flow better given the wide scope of results. However, we appreciate this feedback, and will go back over the section and more clearly demarcate the empirical findings and the related discussion.
>
> We appreciate the feedback regarding our choice of topic modeling methods. We will consider using a different topic model in future work, but given the timeframe, for this paper, we plan to instead consider other approaches to providing more context on the topic and LIWC changes within the extra page provided for the camera-ready.
>
> We will expand on the related work for the camera ready version, and we will release code!

---

> > ### Comment · AnonReviewer2 · 2020-09-28
> > **Reorganisation**
> >
> > You'll note that Reviewer 1 was unable to distinguish between your speculation and your findings as well. Separating the sections is a definitive way to address this very important issue.
> >
> > Appearing to have unsupported findings is a much bigger detriment than having exemplary prose is a good.

---

> > > ### Author Response · Authors · 2020-09-29
> > > **Response to Reorganization**
> > >
> > > Thank you for your thoughtful comment. We will separate them fully for the camera ready (either in distinct sections or by adding subsections for discussion under the question headings).

---

### Official Review · AnonReviewer1 · 2020-09-18
**Interesting paper; some conclusions need to be reviewed**

**Rating:** 6
**Confidence:** 4

**Review:**

This paper describes the analysis of messages published on Reddit under a set of specific subforums related with mental health topics. The authors have performed several analysis to understand the levels of preoccupation of the population on these forums with respect to the pandemia and how the people that use these forums rise. On the other hand a second analysis about the language and topics of discussion has been performed. The topic addressed is interesting and the technical background is solid. My main concern is refered to several points where some conclusions were arisen by the authors with respect the implications of the results obtained. Since the authors didn't present any analysis of the contextualization of the topics or, for example, some sentiment analysis performed over the texts some of these conclusions can be false. In this context, I recommend to check the paper and consider rewriting some of these parts.

---

> ### Author Response · Authors · 2020-09-28
> **Response to Reviewer 1**
>
> Given the extra page for the camera ready, we intend to consider how we can provide more explanatory information about the topics, and provide additional details and analyses for our experiments.

---

### Official Review · AnonReviewer3 · 2020-09-24
**An interesting hypothesis in an important problem, but with a superficial analysis that does not provide a novel understanding of such problem.**

**Rating:** 4
**Confidence:** 3

**Review:**

**Core review:**

This paper considers the hypothesis that the impact of COVID-19 in public mental health (specifically in individuals with mental health issues) can be measured by analyzing the users' behaviour in online forums related to mental health. Specifically, the authors focus on 3 mental-health-related forums in the Reddit social network, measuring the increase or decrease in both the user activities as well as in the relative prevalence of certain keywords of interest. Authors apply time series analysis to evidence a significant difference in user's activity post-COVID with respect to the behaviour predicted by autoregressive models. Furthermore, they analyze the relative change of word usage in certain categories, using LIWC clusters and an LDA topic model.

The main findings indicate that these forums indeed exhibit abnormal behaviour, i.e., it does show an increased activity in these forums that coincides in time and textual content with the COVID-19 pandemic. However, two of the three forums analyzed actually exhibit a decrease in user activity, an issue which is noted by the authors but there is no attempt to explain it. I also consider that the techniques and analysis applied are an initial approximation, but they lack the sophistication expected in a top-level NLP venue. Specifically, the paper relies on descriptive statistics (word counting of LIWC clusters) and topic modelling techniques (LDA) which fail to account for complex linguistic phenomena that may be acting as confounding factors. Is my understanding that the fact that certain keywords show an increased prevalence does not necessarily imply that users are actually feeling an increased mental-health burden. Reaching this conclusion would require a deeper semantic analysis of the text.

As such, I believe the paper establishes a clear and important hypothesis, provides initial evidence for it, and it is clearly written, but the contributions are not sufficient to demonstrate an actual increase in mental health issues in users of these platforms as opposed to an increase in the general discussion of topics related to the COVID-19 pandemic. I commend the authors for their efforts in this clearly important endeavour and I suggest they continue strengthening this analysis.

**Reasons to accept:** The paper is clearly written, the methodology is thoroughly explained and carried out and the problem under discussion is relevant to the workshop.

**Reasons to reject:** I believe the paper does not advance our understanding of the scientific problem under study, that is, the impact of COVID-19 in public mental health, as it does not provide any new insight or result in this matter.

---

> ### Author Response · Authors · 2020-09-28
> **Response to Reviewer 3**
>
> We thank the reviewer for pointing out that our analysis could be improved through the use of more sophisticated linguistic analysis methods. This is something we will consider for future work. We would like to note that both LIWC and LDA topic modeling are well-established tools for linguistic analysis that are frequently used in prior work, especially in the NLP for mental health domain. We believe that their consistent use in prior work validates their efficacy for extracting insights from our data. Further, while we agree that an increase in certain keywords does not guarantee that there was an increased mental health burden, we do believe that by analyzing multiple keywords and contextualizing them within the findings of related work, we can provide useful evidence as to how mental health conversations have evolved since COVID-19. Our goal was not solely to determine if mental health concerns have increased or decreased; instead we also aim to characterize how discussion around mental health concerns have changed, and we believe that our analysis achieves this.
>
> Regarding the decrease in activity within two of the forums, we do not believe that this is necessarily an "issue." As we noted in the paper, this finding did not align with our expectations; however, that does not mean it is invalid. The implications of COVID-19 are still not well understood, and we think part of the value of doing analyses like this one is discovering trends that are not immediately obvious. Note that we have provided some explanation for the decrease, specifically see our discussion of delayed symptom trajectories in depression in section 5.2. We will nonetheless look into additional ways we can better understand this unexpected result for the final version.